# Synthesis of a Series of Dual-Functional Chelated Titanate Bonding Agents and Their Application Performances in Composite Solid Propellants

**DOI:** 10.3390/molecules25225353

**Published:** 2020-11-16

**Authors:** Guomin Lin, Yixue Chang, Yu Chen, Wei Zhang, Yanchun Ye, Yanwen Guo, Shaohua Jin

**Affiliations:** 1Department of Pharmaceutical Botany, School of Material Science and Engineering, Beijing Institute of Technology, Beijing 100081, China; bitmseacm@yeah.net (G.L.); changyixuedhr@163.com (Y.C.); cylsy@163.com (Y.C.); jinshaohua@bit.edu.cn (S.J.); 2Xi’an Modern Chemistry Research Institute, Xi’an 710065, China; 3School of Chemistry & Chemical Engineering, Beijing Institute of Technology, Beijing 100124, China; yeyanchun@sina.com (Y.Y.); guoyanwen@bit.edu.cn (Y.G.)

**Keywords:** bonding agent, chelated titanate, solid propellant, interfacial interaction

## Abstract

Titanate-based bonding agents are a class of efficient bonding agents for improving the mechanical properties of composite solid propellants, a kind of special composite material. However, high solid contents often deteriorate the rheological properties of propellant slurry, which limits the application of bonding agents. To solve this problem, a series of long-chain alkyl chelated titanate binders, *N*-n-octyl-*N*, *N*-dihydroxyethyl-lactic acid-titanate (DLT-8), *N*-n-dodecyl-N, *N*-dihydroxyethyl-lactic acid-titanate (DLT-12), *N*-n-hexadecyl-*N*, *N*-Dihydroxyethyl-lactic acid-titanate (DLT-16), were designed and synthesized in the present work. The infrared absorption spectral changes of solid propellants caused by binder coating and adhesion degrees of the bonding agents on the oxidant surface were determined by micro-infrared microscopy (MIR) and X-ray photoelectron spectroscopy (XPS), respectively, to characterize the interaction properties of the bonding agents with oxidants, ammonium perchlorate (AP) and hexogen (RDX), in solid propellants. The further application tests suggest that the bonding agents can effectively interact with the oxidants and effectively improve the mechanical and rheological properties of the four-component hydroxyl-terminated polybutadiene (HTPB) composite solid propellants containing AP and RDX. The agent with longer bond chain length can improve the rheological properties of the propellant slurry more significantly, and the propellant of the best mechanical properties was obtained with DLT-12, consistent with the conclusion obtained in the interfacial interaction study. Our work has provided a new method for simultaneously improving the processing performance and rheological properties of propellants and offered an important guidance for the bonding agent design.

## 1. Introduction

Solid propellant, a kind of special composite material, is an important energy source for rockets, missiles and weapons [1,2]. The mechanical properties of solid propellants are the key to ensuring the performance stability and service life of the corresponding products, as well as the propellants’ energy performance and stable combustion performance [3]. In a solid propellant, the content of filler, such as AP and RDX, can be up to 50–90%. Therefore, the interfacial bonding between the filler and binder matrix is the key factor affecting the mechanical properties of propellants [4], and bonding agents have been essentially used to improve the interfacial binding for desired mechanical properties [5]. Bonding agents usually contain multifunctional groups, where one end can bind with the solid filler by physical interactions, and the active group participates in the curing reaction of binders to create a bridge between the binder and the solid filler [6].

Since the first application of ethanol amine bonding agents to polyurethane propellants, various types of bonding agents had been developed, such as ethanol amine and its derivatives, aziridine and its derivatives, polyamines and their derivatives, organosilanes, titanates, neutral polymer bonding agents (NPBA), and so on [7,8,9,10,11,12]. Titanate bonding agents are a type of small molecule bonding agents with the general formula of Ti(OR)_4_, where R represents the organic groups capable of hydrolysis. They can react with a small amount of water on the oxidant surface to be partially hydrolyzed to the complexes of titanate that are deposited on the filler surface to form a titanate coating layer [13]. Other active groups of titanates bind with binder matrices. Therefore, titanate bonding agents can bridge the filler and matrix to improve the mechanical properties of a propellant. Such bonding agents have been extensively studied. Allen [14] applied the *n*-butyl titanate (TBA)-based bonding agents to thermoplastic propellants formulated with styrene-diene block copolymers and naphthenic oil as the binder and AP as the oxidant. They found that both high and low temperature mechanical properties of the propellants were greatly improved. It has also been reported [15] that adding 0.1–1.0% of titanate bonding agent in the thermoplastic propellant with styrene-diene copolymer as the binder and AP as the oxidant can enhance the adhesion between thermoplastic elastomer and oxidant, and thereby greatly improve the high, low and normal temperature mechanical properties of the propellant. Monte et al. [16] pointed out that the introduction of organic titanate phosphate could improve the tensile strength and elasticity of a propellant, while enhancing its ballistic performance. The novel alkoxy organic titanate developed by Monte et al. [17] showed excellent performances in improving the uniformity of solid particle dispersion and enhancing the processing and mechanical properties of energetic composite materials.

In recent years, with the development trend of propellant towards high energy and high density, the solid content of propellant has been significantly increased, which inevitably leads to high viscosities and processing performance deteriorations of propellant systems [18]. An optimal process aid can effectively improve the processing performance of propellants [19,20,21]. However, adding two types of inert auxiliaries, the bonding agent and process auxiliary, can significantly reduce the energy performances of the solid propellants [22]. In addition, the improving effects of processing aids on the rheological properties of propellant often conflict with the positive effects of bonding agents on the interfacial interaction. The bonding-processing dual-functional additives capable of improving both bonding and processing performances of propellant have rarely been reported. Designing dual-functional additives that can bring the ideal mechanic properties, processing adjustment and good energy performance to solid propellants simultaneously is urgently needed.

Our study aimed to develop a dual-functional titanate bonding agent that improves both bonding and processing performances of propellant simultaneously. A series of long-chain alkyl chelated titanate bonding agents containing both polar groups and non-polar long-chain alkyl groups were designed and synthesized using tetrabutyl titanate, dihydroxyethylamine, and long-chain alkyl halides as the raw materials and lactic acid as the chelating agent. The polar groups could interact with the solid filler and adsorb on the filler surface to improve the mechanical properties of the propellant. The long-chain alkyl groups were distributed on the outer surface of filler to increase the fluidity of the composite solid propellant slurry for better processing performance. Our work is of great guiding value to explore new methods for developing dual-functional titanate bonding agents that can simultaneously improve the processing and rheological properties of propellants.

## 2. Results and Discussion

### 2.1. NMR Characterization of Bonding Agents

The structure of DLT-8, DLT-12, and DLT-16 were characterized by 1H-NMR spectroscopy. The bonding agents exhibited similar 1H-NMR spectra, with different numbers of carbons in the long alkyl chain (Figure 1). The chemical shift of Ha is at 0.8–1.0 ppm, while those of Hb, Hc and Hd are located in the range of 1.0–1.5 ppm. The number of hydrogen atoms increases with the increase in carbon number; correspondingly, the peak area is increased. The peaks at 3.3–3.7 ppm and 4.0–4.4 ppm can be assigned to Hf and Hg, respectively. The asymmetry of the four split may be derived by diastereotopic protons of the chiral center on the alfa-carbon of the lactic acid ligand. Overall, the regularity of the NMR spectrum was as expected. These results suggested that the target bonding agents with different long-chain alkyl groups had been synthesized.

### 2.2. FTIR Characterization of Bonding Agents

The FTIR spectra of titanyl lactate, DLT-8, DLT-12, DLT-16, and long-chain alkyl intermediates are shown in Figure 2. All three bonding agents exhibited peaks at 1681 cm^−1^ due to the stretching vibration of the carbonyl on ester group. The peaks at 1117 cm^−1^ and 1598 cm^−1^ can be ascribed to the stretching vibration of C–N and the stretching vibration of carbonyl, respectively. These results indicate that a cyclic chelated titanate structure was formed from the hydroxy group via transesterification reaction.

### 2.3. Characterization of Interfacial Interaction by MIR

#### 2.3.1. MIR Analysis of Bonding Agent-Coated AP

The micro-infrared spectra of AP and bonding agents coated with AP samples, AP-DLT-8, AP-DLT-12, and AP-DLT-16, are obtained as shown in Figure 3. AP exhibits three main IR absorption peaks at 3264 cm^−1^, 1405 cm^−1^ and 1072.2 cm^−1^ due to the stretching vibration, deformation vibration and bending vibration of N–H, respectively. The absorption peak of Cl-O in the mid-infrared region is very weak.

The DLT-8, DLT-12, and DLT-16 coatings shift the micro-infrared spectrum of AP to the higher wavelength region by 7.7cm^−1^, 23.2 cm^−1^ and 19.3 cm^−1^, respectively. It can be explained that the lone pair of the carbonyl group in the bonding agent forms a hydrogen bond with the N–H bond of AP [23], as shown in the inserted image. Therefore, the absorption peak caused by the bending vibration of the N–H plane is shifted, while that of the deformation vibration remains unchanged.

The closer the C=O group of the bonding agent is to the N–H bond of AP, the greater the energy required for the in-plane bending vibration of the N–H bond [24]. Based on the shifts of the absorption spectra, it can be concluded that the strengths of the interactions between AP and the bonding agents are in the order of DLT-12 > DLT-16 > DLT-8. It can be explained that the entanglement of bonding agent with AP particles becomes stronger as the alkyl chain length of bonding agent increased, which results in stronger interactions. However, extremely long alkyl chains are highly hydrophobic and show strong steric hindrances, which interfere with the affinity of the titanate group with AP.

#### 2.3.2. MIR Analysis of Bonding Agent-Coated RDX

The micro-infrared spectra of RDX and bonding agent coated RDX samples, RDX-DLT-8, RDX-DLT-12, and RDX-DLT-16 were respectively measured. As shown in Figure 4, RDX exhibits an absorption peak at 1263 cm^−1^ due to the symmetrical stretching vibration of -NO_2_. The DLT-8, DLT-12 and DLT-16 coatings shift the absorption peak of RDX towards the high wavenumber region by 2.2 cm^−1^, 2.7 cm^−1^ and 4 cm^−1^, respectively, due to the charge induction between the ester group in bonding agent and the nitramine group of RDX [25,26], as shown in the inserted image. The ester group on the ring reflects the electron-donating properties as a whole and can coordinate with the nitramine group on RDX. Based on the absorption peak shifts, it can be concluded that the strengths of the charge induction formed between the bonding agents and RDX are in the order of DLT-16 > DLT-12 > DLT-8. RDX is more hydrophobic than AP, and therefore its hydrophobic repulsion causes the ester group to be exposed after the bonding agent is coated with RDX, participating in the interaction with the nitramine group. The molecular chain of DLT-16 is the longest, and thus the interaction between its ester group with the nitramine group of RDX is the strongest.

### 2.4. Quantitative Analysis of Interfacial Interaction by XPS

XPS is a powerful tool for the characterization of surface-acting properties [27]. Here, we quantitatively analyzed the interaction between the bonding agents and oxidants by XPS. The MIR analysis suggested that both AP and RDX interact with the bonding agents mainly via the N atom. To avoid the interference of carbon contaminations [28], the adhesion degrees of the bonding agents on AP and RDX surfaces were evaluated with the percentage of the 1s peak intensities of different N atoms over the total peak intensity of N in 1s photoelectron spectra obtained by narrow scan using the equation below.
(1)$$R=\frac{A_{N1S-B}}{A_{N1S-C}}$$
where *A_N_*_1s–B_ is the peak area of >N- and -CN in the narrow scan and *A_N_*_1s–C_ refers to total peak area of N1s in the narrow scan. The calculation results are shown in Table 1. Figure 5 shows the fitting curve of the N1s XPS spectrum of each sample.

As can be seen from Table 1 and Figure 5, all of the AP and RDX samples coated with the bonding agents showed the N1s XPS peaks of >N from the bonding agents. The interaction of the oxidants with bonding agents decreases the electron density of -NO_2_, and thus shifts the emission peak to the high binding energy region. The adhesions of DLT-8, DLT-12, and DLT-16 on the AP surface were determined to be 42.50%, 55.07% and 51.08%, respectively, and those on RDX surface were found to be 4.82%, 6.72% and 11.49%, respectively. The changes in the adhesion degrees of the bonding agents on the AP and RDX surfaces with alkyl chain length obtained by XPS analysis is consistent with the MIR results. However, the quantitative analysis with the narrow-scan N1s XPS peaks of different states can better reflect the interaction strength between bonding agent and oxidants.

### 2.5. Practical Application of Chelated Titanate Bonding Agents to Hydroxyl-Terminated Polybutadiene (HTPB) Composite Propellants

The application potentials and bonding performances of the chelated titanate bonding agents were evaluated in the four-component HTPB composite propellants. The HTPB composite propellants formatted with the chelated titanate bonding agents were tested for their mechanical properties at high, normal and low temperatures. An HTPB composite propellant was also formulated with the traditional BA-12 bonding agent for comparison purpose. As shown in Table 2, the chelated titanate bonding agents with long alkyl chains significantly reduced the yield value of the HTPB propellant slurry. In addition, the longer the alkyl chain of the chelated titanate bonding agent, the lower the yield value of the propellant slurry was [29]. Therefore, the introduction of bonding agents with long-chain alkyl groups can improve the processing performance of propellant slurry.

Mixing the chelated titanate bonding agent with four-component HTPB propellant significantly increased maximum tensile strength (σ) at high, normal and low temperatures, reduced the adhesion index *Φ*, and slightly decreased the elongation under maximum tensile strength (*ε_m_*) and the elongation at break (*ε_b_*) of the propellant. It can also be derived from Table 2 that the HTPB propellants using the above bonding agent showed better tensile strength at high temperature and good elongation at low temperature. These results indicate that the chelated titanate bonding agents can effectively improve the interfacial interactions between the solid component and binder matrix. In addition, DLT-12 exhibited the best performance in improving the mechanical properties of the four-component HTPB propellant. The long alkyl chain of DLT-16 can interfere with its binding effect on the solid particles and binders to certain degrees. In addition, the AP content in the HTPB propellant is very high. Therefore, the interaction between bonding agent and AP had a greater influence on the mechanical properties of propellant. As demonstrated above, DLT-12 had the strongest coating effect on AP, and thus it exhibited the best bonding performance in the propellant. Taking these comprehensive effects of the chelated titanate bonding agents on the processing and mechanical properties of the propellant into consideration, DLT-12 was determined to be an optimal bonding agent for HTPB composite propellants.

## 3. Experimental Section

### 3.1. Materials

Analytical grade bromo-n-octane, bromododecane, and bromohexadecane were provided by Sinopharm Chemical Reagent Co., Ltd. (Beijing, China). Industrial grade ammonium perchlorate (AP) and hexogen (RDX) were purified by passing them through sieves of 200 mesh (0.074 mm) and 300 mesh (0.048 mm), respectively, and dried at 60 °C before use. Racemic D, L-lactic acid provided by Aladdin Reagent (Shanghai, China) was dried over MgSO_4_ and filtered. All other reagents (analytical grade) were purchased from Beijing Chemical Plant (Beijing, China) and used as received.

### 3.2. Synthesis of N-Long Chain Alkyl-N, N-Dihydroxyethylamine

#### 3.2.1. Synthesis of *N*-Octyl-*N*, *N*-Dihydroxyethylamine

To a solution of dihydroxyethylamine (15.75 g, 0.15 mol) in 50 mL of alcohol, octyl bromide (30.5 g, 0.1 mol) and NaOH (4 g, 0.1 mol) were added. The mixture was refluxed at 80–90 °C under constant stirring for 1h, cooled to room temperature, washed with water to remove NaBr and alcohol, and vacuum dried to produce *N*-octyl-*N*, *N*-dihydroxyethylamine as a colorless and oily liquid with a yield of 95%. FTIR (KBr, cm^−1^): 3377(ν_OH_), 2928(ν_asCH_), 2857(ν_sCH_), 1462(δ_CH_), 1373(δ_OH_), 1114(ν_C-N_), 1063(ν_C-O_), 876(δ_C-O_),723(ρ_CH2_). ^1^H-NMR(DMSO, 300Hz): δ 0.88 (br, a-H, 3H), 1.29 (br, b-H, c-H, 10H), 1.34–1.39 (br, d-H, i-H, 5H), 2.33 (br, e-H, 2H), 3.30 (br, f-H, 4H), 4.10–4.20 (br, g-H, h-H, 5H).

#### 3.2.2. Synthesis of *N*-Dodecyl-*N*, *N*-Dihydroxyethylamine

*N*-dodecyl-*N*, *N*-dihydroxyethylamine was prepared using the same apparatus and procedures as described in Section 3.2.1, except that octyl bromide was replaced with dodecyl bromide. The product of *N*-dodecyl-*N*, *N*-dihydroxyethylamine was obtained as a colorless and oily liquid with a yield of 94%. FTIR (KBr, cm^−1^): 3388(ν_OH_), 2925(ν_asCH_), 2855(ν_sCH_), 1463(δ_CH_), 1372(δ_OH_), 1115(ν_C-N_), 1060(ν_C-O_), 874(δ_C-O_), 722(ρ_CH2_). ^1^H-NMR(DMSO, 300Hz): δ 0.88 (br, a-H, 3H), 1.29 (br, b-H, c-H, 18H), 1.34–1.39 (br, d-H, i-H, 5H), 2.33(br, e-H, 2H), 3.30 (br, f-H, 4H), 4.10–4.20 (br, g-H, h-H, 5H).

#### 3.2.3. Synthesis of *N*-Hexadecyl-*N*, *N*-Dihydroxyethylamine

Similarly, *N*-hexadecyl-*N*, *N*-dihydroxyethylamine was prepared using the same apparatus and procedures as described Section 3.2.1 with octyl bromide substituted with hexadecyl bromide. The product of *N*-hexadecyl-*N*, *N*-dihydroxyethylamine was obtained as a colorless and oily liquid with a yield of 95%. FTIR(KBr, cm^−1^): 3377(ν_OH_), 2928(ν_asCH_), 2857(ν_sCH_), 1462(δ_CH_), 1373(δ_OH_), 1114(ν_C-N_), 1063(ν_C-O_), 876(δ_C-O_), 723(ρ_CH2_). ^1^H-NMR(DMSO, 300Hz): δ 0.88 (br, a-H, 3H), 1.29 (br, b-H, c-H, 26H), 1.34–1.39 (br, d-H, i-H, 5H), 2.33 (br, e-H, 2H), 3.30 (br, f-H, 4H), 4.10–4.20 (br, g-H, h-H, 5H).

### 3.3. Synthesis of Lactic Acid Dibutoxytitannate Ester

To a solution of tetrabutyl (28.2 g, 0.1 mol) in 50 mL of 1-butanol, lactic acid (34.0 g, 0.1 mol) was added dropwise under constant stirring. The mixture was allowed to react at 80 °C under vacuum distillation conditions (1.33 × 10^4^ Pa Vacuum) for 1–2 h until the mixture became a solid powder. The solid powder was vacuum dried to produce lactic acid-dibutoxytitannate ester as a yellow solid powder with a yield of 94%. FTIR (KBr, cm^−1^): 2959(ν_asCH_), 2933(ν_asCH_), 2870(ν_sCH_), 1700(ν_C=O_), 1460(δ_asCH_), 1419(δ_asCH_), 1367(δ_sCH_), 1306(δ_CH_), 1259(ν_C-O_), 1124(ν_asC-O_), 1092(ν_sC-O_).

### 3.4. Synthesis of Conjugated Titanate Bonding Agents

The three long chain intermediates respectively reacted with dibutyl titanate lactate to produce the corresponding titanate bonding agents with different chain lengths. For a typical synthesis process, lactic acid-dibutoxytitannate ester (0.05 mol) was added into a solution of *N*-octyl-*N*, *N*-dihydroxyethylamine (0.05 mol) in 50 mL 1-butanol. The mixture was allowed to react under constant stirring and vacuum distillation conditions (1.33 × 10^4^ Pa Vacuum) at 80 °C for 2 h to produce a light yellow waxy solid at room temperature. The waxy solid was smashed and vacuum dried to produce DLT-8 as a light-yellow solid powder. DLT-12 and DLT-16 were synthesized with the same equipment and procedure using *N*-dodecyl-*N*, *N*-dihydroxyethylamine and *N*-hexadecyl-*N*, *N*-dihydroxyethylamine, respectively, instead of *N*-octyl-*N*, *N*-dihydroxyethylamine. Figure 6 shows the synthesis route.

### 3.5. Characterization of Chelated Titanate Binding Agents

*^1^H-NMR.*^1^H-NMR spectra were measured with an ARX300 nuclear magnetic resonance spectroscope (Bruker, Swiss) using deuterated DMSO as the solvent (H = 2.50 ppm) at a frequency of 300 MHz. The spectral width was 130 KHz. Inversion-gated decoupling was applied with a pulse of 90° and repetition period of 7s. The nuclear magnetic resonance spectrum was accumulated more than 7000 times.

*FTIR.* A small amount of synthesized product was mixed with KBr power, pressed into KBr pellets, and analyzed with a NEXUS-470 FTIR spectrometer (Madison, American) for IR spectrum.

### 3.6. Interfacial Interaction Between Chelated Titanate Bonding Agents and AP/RDX

The interaction between bonding agents and solid filler has been a prominent research focus. Understanding such interactions can not only evaluate the role of the bonding agent, but also provide a theoretical basis for the further developing and screening of bonding agents. For this purpose, AP and RDX were coated with the bonding agents, DLT-8, DLT-12, and DLT-16, respectively, by a hybrid coating method. The interactions between these bonding agents and oxidants were characterized by micro-infrared spectroscopy (MIR) and X-ray photoelectron spectroscopy (XPS).

For a typical preparation procedure, 0.5 g of bonding agent was completely dissolved in 10 mL ethanol, mixed with 1.0 g of AP, stirred at room temperature for 1 h, and filtrated. The solid residue was washed with ethanol for three times and dried in an oven at 60 °C for 2 days to remove the moisture and solvent, which produced the bonding agent coated oxidant.

MIR analysis was conducted on a Nicolet Magna-IR750 Fourier transform infrared spectrometer using the microscope infrared accessory and KBr pellets. Spectra were recorded at the resolution of 3 cm^−1^ with 200 accumulated scans.

XPS spectra were recorded on a Perkin-Elmer Φ5300 X-ray photoelectron spectrometer using the MgKa X-ray source (hv = 1486.6 eV) with a power of 250 W. XPS data were processed in the proprietary software XPS-PEAK by curve smoothing, deconvolution, X-ray satellite line deduction, background deduction, normalization, and curve fitting.

### 3.7. Application of Bonding Agents to HTPB Composite Propellants

Four-component HTPB composite solid propellants (HTPB/AP/RDX/Al) were formulated with the prepared bonding agents, respectively, and their mechanical properties were characterized. The propellant was composed of 83 wt.% total solid (42 wt.% AP, 24 wt.% RDX and 17 wt.% Al powder), 16.8 wt.% HTPB binder system, and 0.2 wt.% bonding agent. The amide bonding agent BA-12, commonly used in HTPB propellants, was used as a control. The *R* value of the propellant binder system was 0.98. The propellant slurry was kneaded and cured at 60 °C for 7 days after pouring. The cured sample was sliced, cut into dumbbell-shaped pieces following the ASTM1708-95 standard, and tested with an Instron universal testing machine (Instron 1185, Instron Co., Ltd., High Wycombe, UK), at −40 °C, 25 °C, and 60 °C, respectively. The drawing speed was set to 100 mm·min^−1^. The maximum tensile strength (*σ_m_*), elongation under maximum tensile strength (*ε_m_*), and elongation at break of the propellant (*ε_b_*) were obtained from the stress-strain curves.

## 4. Conclusions

A series of dual-functional chelated titanate bonding agents, DLT-8, DLT-12, and DLT-16, with different alkyl chain lengths were prepared and their properties were characterized. The interfacial interactions of the bonding agents with AP and RDX were studied by MIR and XPS. The results revealed that all three bonding agents can strongly adhere onto both AP and RDX, with the bonding strengths in the order of DLT-12 > DLT-16 > DLT-8 on the surface of AP and in the order of DLT-16 > DLT-12 > DLT-8 on the surface of RDX. The practical application potentials of the bonding agents were evaluated in the four-component HTPB composite propellant. It was found that the introduction of a long alkyl chain to the bonding agent can effectively improve the processing performance of propellants. All three chelated titanate bonding agents with different alkyl chain lengths improved the mechanical properties of the propellant. DLT-12 exhibited the best comprehensive effects on both processing performance and mechanical properties of propellant slurry. These long alkyl chain chelated titanate bonding agents exhibited dual functions to improve both mechanical and processing properties of propellants, which solved the problems encountered by the traditional titanate bonding agents which cannot improve the processing properties of propellants. The novel dual-function titanate bonding agents can effectively reduce the use of inert auxiliaries in propellants, and are of significant value for improving the mechanical, processing and energy properties of propellants.

## Figures and Tables

**Figure 1 molecules-25-05353-f001:**
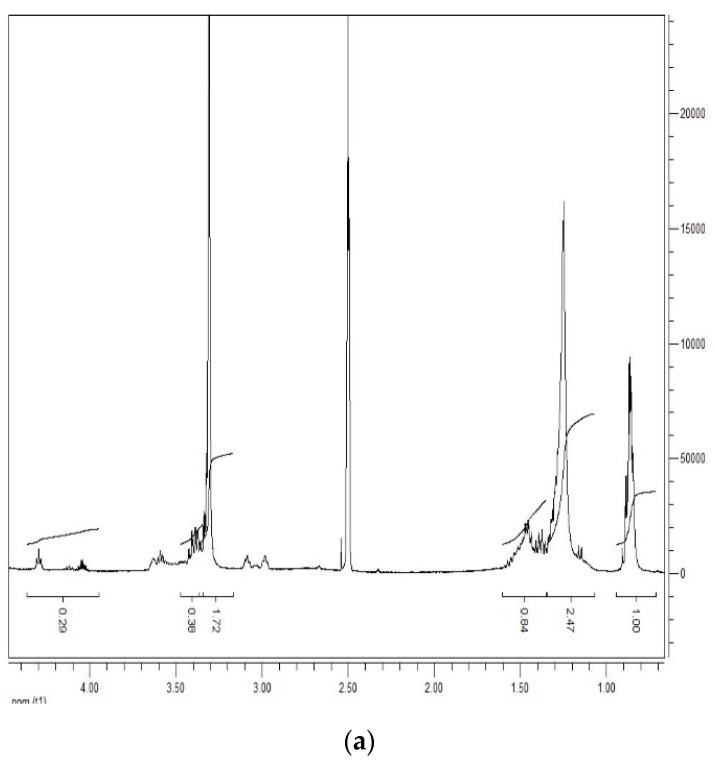
^1^H-NMR spectra of DLT-8 (**a**), DLT-12 (**b**), DLT-16 (**c**) and 1H-NMR labeling of product molecules (**d**).

**Figure 2 molecules-25-05353-f002:**
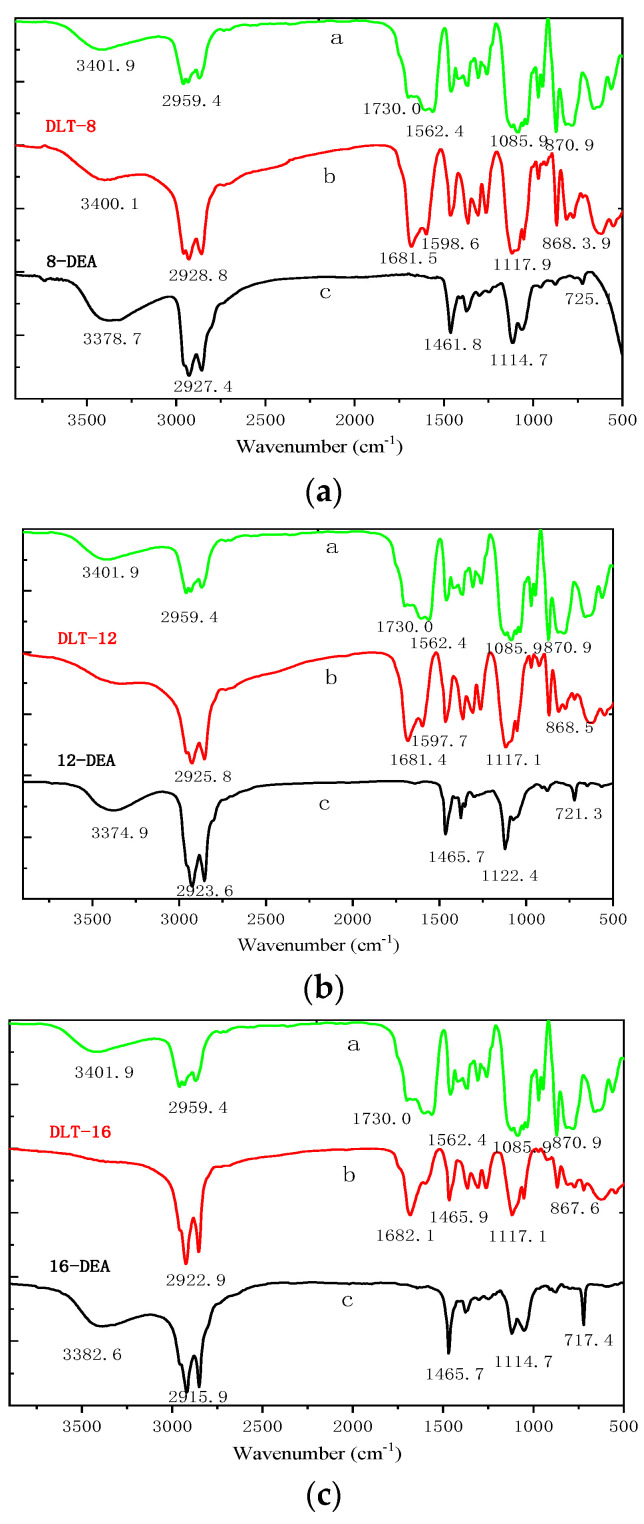
FTIR spectra of lactate titanate (**a**), DLT-8, DLT-12, and DLT-16 (**b**), and long chain alkyl intermediate (**c**).

**Figure 3 molecules-25-05353-f003:**
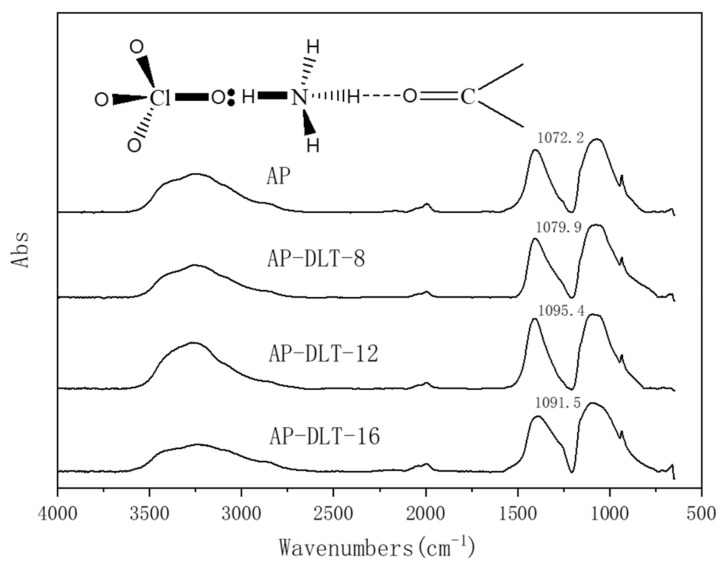
Micro-infrared spectra of ammonium perchlorate (AP) and AP coated with DLT-8, DLT-12 and DLT-16, and the interaction mechanism between the bonding agents and AP (inserted image).

**Figure 4 molecules-25-05353-f004:**
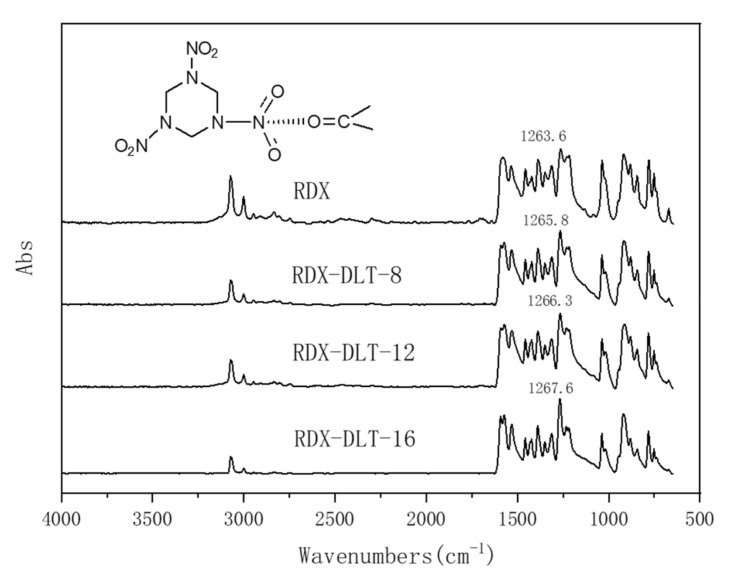
Micro-infrared spectra of hexogen (RDX) and RDX coated with DLT-8, DLT-12 and DLT-16, and the interaction mechanism between the bonding agents and RDX (inserted image).

**Figure 5 molecules-25-05353-f005:**
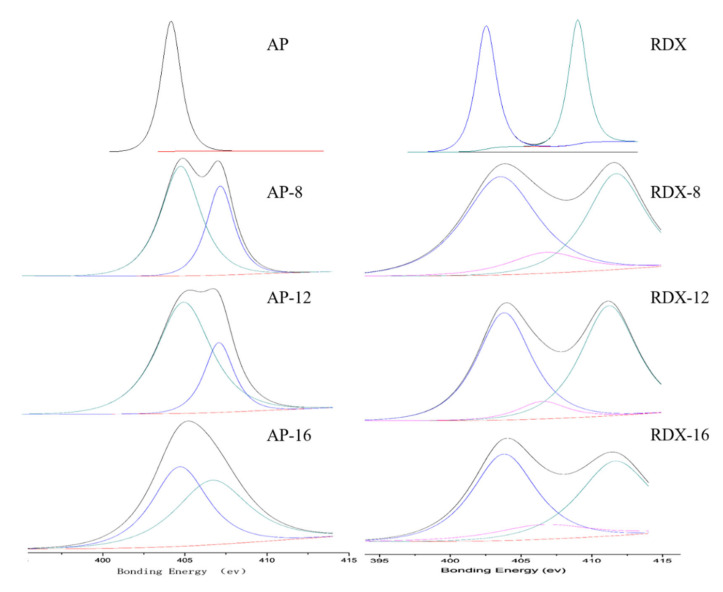
Fitting curves of the XPS N1s spectra of pure AP and RDX solid particles and the particles coated with bonding agents.

**Figure 6 molecules-25-05353-f006:**
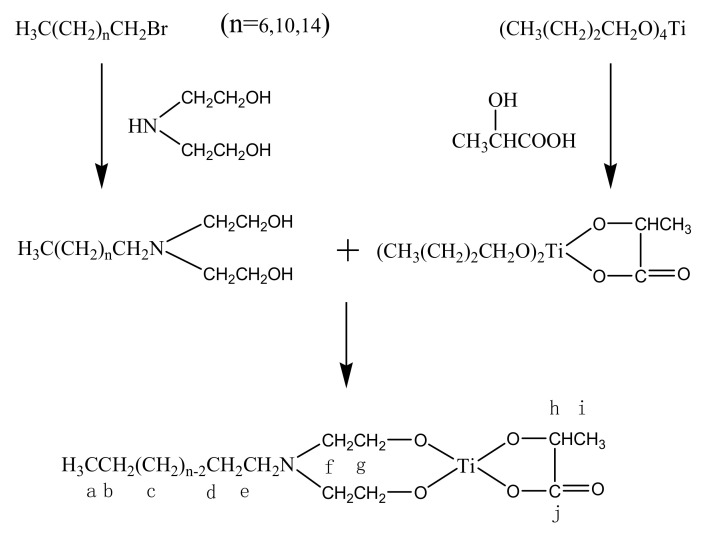
Synthesis route to conjugated titanate bonding agents.

**Table 1 molecules-25-05353-t001:** Specifications and contents of sample N1s spectra.

Sample	Functionality of N Atom	Binding Energy (eV)	Content (%)	R (%)
AP	-NH_4_^+^	404.45	100	0
AP-8	-NH_4_^+^	404.38	57.50	42.50
	>N-	406.66	42.50	
AP-12	-NH_4_^+^	404.94	44.93	55.07
	>N-	406.83	55.07	
AP-16	-NH_4_^+^	404.70	48.92	51.08
	>N-	406.78	51.08	
RDX	-NO_2_	404.06	52.49	0
	>N-NO_2_	409.67	47.51	
RDX-8	-NO_2_	403.90	54.02	4.82
	>N-NO_2_	411.09	41.16	
	>N-	406.64	4.82	
RDX-12	-NO_2_	404.28	45.01	6.72
	>N-NO_2_	410.59	48.27	
	>N-	406. 41	6.72	
RDX-16	-NO_2_	403.70	44.12	11.49
	>N-NO_2_	411.06	44.49	
	>N-	406.51	11.49	

**Table 2 molecules-25-05353-t002:** Effects of the chelated titanate bonding agents on the mechanical properties of HTPB composite propellant.

		Control	DLT-8	DLT-12	DLT-16
**Yield Value/Pa**		39.26 ± 1.04	33.15 ± 0.98	23.71 ± 1.17	18.64 ± 0.89
−40 °C	*σ*/MPa	1.01 ± 0.09	1.61 ± 0.11	1.83 ± 0.19	1.54 ± 0.20
	*ε_m_*/%	37.6 ± 1.12	29.5 ± 1.65	31.6 ± 1.99	28.7 ± 2.47
	*ε_b_*/%	48.9 ± 2.37	41.7 ± 2.19	38.9 ± 1.95	34.7 ± 1.89
	*Φ*	1.30	1.41	1.23	1.21
25 °C	*σ*/MPa	0.47 ± 0.09	0.53 ± 0.12	0.56 ± 0.07	0.54 ± 0.09
	*ε_m_*/%	43.1 ± 2.13	44.9 ± 3.17	41.9 ± 2.09	41.7 ± 2.57
	*ε_b_*/%	66.5 ± 3.55	65.7 ± 3.69	62.1 ± 3.12	60.5 ± 2.99
	*Φ*	1.54	1.46	1.48	1.45
60 °C	*σ*/MPa	0.43 ± 0.08	0.49 ± 0.07	0.67 ± 0.11	0.63 ± 0.13
	*ε_m_*/%	46.3 ± 3.62	43.1 ± 2.55	42.3 ± 2.49	40.5 ± 2.55
	*ε_b_*/%	69.7 ± 3.19	61.6 ± 3.08	59.5 ± 3.17	56.7 ± 3.98
	*Φ*	1.51	1.43	1.41	1.40

*Φ = ε_b_*/*ε_m_*, indicates the adhesion index between filling particles and binder matrix [26].

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
