# Peer review of "Synthesis of a Series of Dual-Functional Chelated Titanate Bonding Agents and Their Application Performances in Composite Solid Propellants"

_molecules, 2020, doi:10.3390/molecules25225353_

Round 1
Reviewer 1 Report
I have no significant comments on the manuscript.
Presenting the conclusions in points would improve the readability of this paragraph, but it is not necessary.
Author Response
Many thanks for the advices of the reviewer.
Reviewer 2 Report
This is an interesting article. The authors very clearly determined the aim of the study. A series of dual-functional chelated titanate bonding agents, DLT-8, DLT-12 and DLT-16, with different alkyl chain lengths were characterized. Synthesis of N-long chain alkyl-N, N-dihydroxyethylamine have been described in detail. The paper is well written - clear motivation, explanations. Figures are very good quality.
However, before allowing for publication, I would suggest making some minor improvements:
1) For DLT-12, the increase in strength compared to the control sample (BA-12) is 19.1% for 25°C, 55.8% for 60°C and 81.2% for -40°C. However, there is no information on how the mechanical properties will change after a certain time? The aging phenomenon is very important. Are further studies planned?
2) Is it possible to compare the cost of the technology for novel dual-function titanate bonding agents and traditional BA-12 ?
3) How many samples in each series were tested in mechanical testing?
4) I suggest adding a book to the literature “Energetic Materials Research, Applications, and New Technologies”
After completing the above mentioned comments, I strongly recommend the paper for publication.
Author Response
Thanks for the scientific and earnest advices prompted to my manuscript by the reviewer and editor. All the revised paragraphs have been highlighted in red color. A list of response to the editor’s comments is attached.
1) For DLT-12, the increase in strength compared to the control sample (BA-12) is 19.1% for 25°C, 55.8% for 60°C and 81.2% for -40°C. However, there is no information on how the mechanical properties will change after a certain time? The aging phenomenon is very important. Are further studies planned?
Thanks for the advice of the reviewer. The aging phenomenon is very important as you said. We are studying the aging phenomenon of the propellant using the above-mentioned bonding agents, and the related research will be introduced in a later article.
2) Is it possible to compare the cost of the technology for novel dual-function titanate bonding agents and traditional BA-12 ?
It is understood that BA-12 is 580 Chinese yuan per kilogram, and the cost of our bonding agents is 650 Chinese yuan per kilogram, which is slightly more expensive than BA-12. But after adding our bonding agents, there is no need to add additional process aids. Therefore, in terms of cost reduction and energy decreasing of inert additives, our dual-function titanate bonding agents are better.
3) How many samples in each series were tested in mechanical testing?
In order to reduce the deviation, we took five samples for parallel testing in each series and took the average value as the result.
4) I suggest adding a book to the literature “Energetic Materials Research, Applications, and New Technologies”
Thanks for the advice of the reviewer. The above book is very important to us and we added it to the literature.
Reviewer 3 Report
Review of paper no. molecules-934512-v1 titled Synthesis of a series of dual-functional chelated titanate bonding agents and their application performances in composite solid propellants by G. Lin et al.
This papers reports the synthesis and characterization of three long-chain alkyl chelated titanate binders for the application as composite solid propellants. The paper is interesting and provides new results. Nevertheless, the compounds are not sufficiently characterized. The paper requires a major revision before being re-considered for possible publication in MDPI-Molecules.
Specific comments:
1.The new compounds (DLT-8, DLT-12, DLT-16) differ in the length of alkyl chains only. As such, it is difficult to distinguish them by 1H NMR and IR only. It is advised to include mass spectra to confirm the different molecular mass.
2.The 13C NMR is also recommended to distinguish the different carbon structure of the molecules.
3.It is not explained why the ligands were chelated to titanium. Is there a specific reason for Ti? What about other metallic cations (Fe, Co, Ni, etc.)?
4.Captions of Figs. 1 and 2 should be interchanged.
5.The experimental stress-strain curves should be included for the sake of comparison.
6.The lowest temperature should be presented first in Table 2.
7.The effect of temperature on tensile strength should be discussed.
8.All abbreviations should be explained at their first mention in the text (RDX, HTPB, etc.).
End of comments
Author Response
Thanks for the scientific and earnest advices prompted to my manuscript by the reviewer and editor. All the revised paragraphs have been highlighted in red color. A list of response to the editor’s comments is attached.
1.The new compounds (DLT-8, DLT-12, DLT-16) differ in the length of alkyl chains only. As such, it is difficult to distinguish them by 1H NMR and IR only. It is advised to include mass spectra to confirm the different molecular mass.
Thanks for the advice of the reviewer. One part of the article explains that we have synthesized this series of bonding agents, and the second part, as the key content, compares and studies its interaction with AP and RDX. We believe that IR and 1H-NMR have been able to fully explain the synthesis of the target product.
2.The 13C NMR is also recommended to distinguish the different carbon structure of the molecules.
Thanks for the advice of the reviewer. As mentioned in the answer to the first question, it is proved that we have obtained the target product by 1H NMR and IR. We focus on the comparison of the application rules of the target product.
By the way, the peak area of Hb, Hc and Hd are increased as increasing of carbon number, which can be easily observed in 1H NMR result. So we think that the above obesrvation is enough to reveal the differ in the length of alkyl chains
3.It is not explained why the ligands were chelated to titanium. Is there a specific reason for Ti? What about other metallic cations (Fe, Co, Ni, etc.)?
As a commonly used organometallic compound, tetrabutyl titanate easily reacts with lactic acid to form a structure with both a strong polar cyclic part and a weakly polar branched part, which has the potential to develop into a type of bonding agent with excellent properties. The references below also support this opinion.
(Li, H.; Deng, J.; Tang, H. Chelated titanate aids used in nitrate plasticized polyether propellant, Journal of Propulsion Technology. 2000, 4, 73-76.)
Besides, tetravalent titanium is relatively stable, and there are empty orbitals that can bond with alkyl ligands to form a stable chelate.
4.Captions of Figs. 1 and 2 should be interchanged.
Thanks for the advice of the reviewer. We have considered your comments and made adjustments in the revision.
5.The experimental stress-strain curves should be included for the sake of comparison.
Thanks for the advice of the reviewer. Since the experimental results are averaged, if the stress-strain curve is added, it will be a bit messy, so we did not add this part after consideration.
6.The lowest temperature should be presented first in Table 2.
Thanks for the advice of the reviewer. The lowest temperature have been presented first in Table 2.
7.The effect of temperature on tensile strength should be discussed.
Thanks for the advice of the reviewer. The effect of temperature on tensile strength have been discussed in the revision.
8.All abbreviations should be explained at their first mention in the text (RDX, HTPB, etc.).
Thanks for the advice of the reviewer. We have checked to make sure that all abbreviations have been explained at their first mention in the text.
Round 2
Reviewer 3 Report
Review of paper no. molecules-934512-v2 titled Synthesis of a series of dual-functional chelated titanate bonding agents and their application performances in composite solid propellants by G. Lin et al.
This is a revised version of a previously reviewed paper. Although some comments have been answered, the paper still requires a major revision. The following issues have not been resolved:
1.The compounds are not sufficiently characterized. Fig. 1 is a low resolution 1H NMR. The alkyl peaks at 1.20-1.30 ppm overlap one another. Furthermore, DLT-12 has a more intense signal at 1.20-1.30 ppm compared to DLT-16. As such, the peak intensity does not correspond to the expected number of protons. It is recommended to either include high resolution 1N NMR data to clearly distinguish the different protons in the molecule or provide mass spectra to confirm the different molecular mass. The 13C NMR could be also helpful to distinguish the different carbon structure of the molecules.
2.The effect of temperature on tensile strength (σ) of DLT-12 and DLT-16 chelated titanate bonding agents is not linear (Table 2). The non-linear dependence must be properly discussed. The tensile strength of composite propellants should be compared with previously studied materials.
3.If the data presented in Table 2 are an average from several independent measurements, error bars must be included.
End of comments
Author Response
Dear Editor:
We have revised our manuscript,entitled “Synthesis of a series of dual-functional chelated titanate bonding agents and their application performances in composite solid propellants” (manuscript ID: molecules-934512) carefully according to the reviewers’ comments. Thanks for the scientific and earnest advices prompted to my manuscript by you and the reviewers. All the revised paragraphs have been highlighted in red color. A list of response to the editor’s comments is attached. We have resubmitted the manuscript on your website. We are grateful to you, who made great contribution to improve our paper. If there are any more comments on our paper, please let us know.
Yours sincerely
Chen Yu
Answer to the reviewers' comments:
1.The compounds are not sufficiently characterized. Fig. 1 is a low resolution 1H NMR. The alkyl peaks at 1.20-1.30 ppm overlap one another. Furthermore, DLT-12 has a more intense signal at 1.20-1.30 ppm compared to DLT-16. As such, the peak intensity does not correspond to the expected number of protons. It is recommended to either include high resolution 1N NMR data to clearly distinguish the different protons in the molecule or provide mass spectra to confirm the different molecular mass. The 13C NMR could be also helpful to distinguish the different carbon structure of the molecules.
Thanks for the advices of the reviewer. As the purpose of NMR and IR is to show that cyclic chelated titanate structure was formed from the hydroxy group via transesterification reaction. Because the feeding is sure, the hydrogen in the alkyl chain in the NMR is not the main concerns for characterization.
2.The effect of temperature on tensile strength (σ) of DLT-12 and DLT-16 chelated titanate bonding agents is not linear (Table 2). The non-linear dependence must be properly discussed. The tensile strength of composite propellants should be compared with previously studied materials.
Thanks for the advices of the reviewer. We supplied the discussion in section 3.5 to make the discussion for the above results clearer. As the AP content in the HTPB propellant is very high, the interaction between bonding agent and AP has a greater influence on the mechanical properties of propellant.
As demonstrated by the results of MIR and XPS, the strengths of the interactions between AP and the bonding agents are in the order of DLT-12>DLT-16>DLT-8. Moreover, we discussed the reason in section 2.3.1. The entanglement of bonding agent with AP particles becomes stronger as the alkyl chain length of bonding agent increased, which results in stronger interactions. However, extremely long alkyl chains are highly hydrophobic and show strong steric hindrances, which interfer with the affinity of the titanate group with AP.
By the way, the amide bonding agent BA-12 commonly used in HTPB propellants was used as a control. And then the processing and mechanical properties of the propellant was compared.
- If the data presented in Table 2 are an average from several independent measurements, error bars must be included.
Thanks for the advice of the reviewer. The error bars have been supplied in Table 2, and the format of the table is reformatted.
